# Advances in Whole Genome Sequencing: Methods, Tools, and Applications in Population Genomics

**DOI:** 10.3390/ijms26010372

**Published:** 2025-01-04

**Authors:** Ying Lu, Mengfei Li, Zhendong Gao, Hongming Ma, Yuqing Chong, Jieyun Hong, Jiao Wu, Dongwang Wu, Dongmei Xi, Weidong Deng

**Affiliations:** 1Yunnan Provincial Key Laboratory of Animal Nutrition and Feed, Faculty of Animal Science and Technology, Yunnan Agricultural University, Kunming 650201, China; yinglu_1998@163.com (Y.L.); mfli_2000@163.com (M.L.); zander_gao@163.com (Z.G.); mhm1283403564@163.com (H.M.); 2022004@ynau.edu.cn (Y.C.); hongjieyun@163.com (J.H.); 15229238680@163.com (J.W.); danwey@163.com (D.W.); 2State Key Laboratory for Conservation and Utilization of Bio-Resource in Yunnan, Kunming 650201, China

**Keywords:** whole genome sequencing, population genomics, Loci, methods, application

## Abstract

With the rapid advancement of high-throughput sequencing technologies, whole genome sequencing (WGS) has emerged as a crucial tool for studying genetic variation and population structure. Utilizing population genomics tools to analyze resequencing data allows for the effective integration of selection signals with population history, precise estimation of effective population size, historical population trends, and structural insights, along with the identification of specific genetic loci and variations. This paper reviews current whole genome sequencing technologies, detailing primary research methods, relevant software, and their advantages and limitations within population genomics. The goal is to examine the application and progress of resequencing technologies in this field and to consider future developments, including deep learning models and machine learning algorithms, which promise to enhance analytical methodologies and drive further advancements in population genomics.

## 1. Introduction

A genome is the collection of all genetic information of an organism, including all genes and their regulatory and non-coding regions [1]. Genome sequencing reveals the genetic diversity of species, population structure, domestication processes, and evolutionary and adaptive mechanisms, and identifies the loci of variation associated with economically important traits [2,3]. With ongoing advances in high-throughput sequencing technologies and declining costs, a growing number of species have been sequenced to produce high-quality reference genomes. The scope of sequenced organisms has expanded from traditional model organisms to include economically important crops, livestock, and endangered plants, with successful assemblies achieved for species such as yellow cattle [4], yaks [5], zebu [6], goats [7], and sheep [8]. Whole genome sequencing (WGS) refers to sequencing the entire genome of multiple individuals within a species using a known reference genome sequence [9]; this technology enables the precise reading and analysis of the entire DNA sequence of an organism [1]. WGS projects have markedly accelerated the generation of genomic data across various taxa, comprehensively characterizing genetic variation in populations and facilitating the rapid development of population genomics to elucidate biological processes such as natural selection, adaptation, and gene flow. By analyzing genomic variation within and between populations, researchers can identify adaptive traits that confer survival advantages in specific environments [10]. Furthermore, WGS provides insights into the domestication history of species, revealing genetic bottlenecks and the selection pressures that have shaped their genomes over time [11]. Further, information such as population size, mobility, and phylogenetic relationships can be estimated more precisely, improving the accuracy of inferences about effective population size and evolutionary history [12,13]. Broadly, population genomics is divided into two main categories based on research objectives and analysis strategies: the inference of population genetic structure and the detection of genomic variation and selection signals [14].

In recent years, challenges have remained in efficiently processing large-scale sequencing data and effectively integrating population genomics approaches into broader biological studies. The exponential growth of sequencing data has ushered in a new era of genomic research, requiring powerful methods and analysis tools to handle increasingly complex datasets. This review summarizes current methods and key analytical software related to whole-genome sequencing and their various applications in population genomics, discussing the strengths and limitations of each approach. The aim is to provide theoretical references for population inference and evolutionary studies, especially in economically important crops and livestock, and to highlight emerging trends and future directions in the field.

## 2. Overview of Whole Genome Sequencing and Its Methods

### 2.1. Definition of Whole Genome Sequencing

Whole genome sequencing (WGS) and Reduced-Representation Genome Sequencing (RRGS) are both widely used techniques for obtaining genetic information to study genomic variation across species. RRGS was first proposed by Miller and refers to sequencing specific regions or representative fragments of an organism [15]. However, due to its limited genome coverage, RRGS has largely been replaced by WGS, which is more efficient in identifying novel loci. By aligning sequences to known references, WGS identifies various genetic variants such as Single Nucleotide Polymorphisms (SNPs), Structural Variations (SVs), Insertions and Deletions (InDels), and Copy Number Variations (CNVs) [16,17,18,19], providing comprehensive genetic data for exploring inter-individual and inter-population differences.

WGS addresses key evolutionary biology questions that traditional methods, such as Sanger sequencing and PCR amplification, have not fully resolved. Based on sequencing depth, coverage, and sample size, WGS techniques can be categorized into [19] individual sequencing with a high depth of coverage based on differences in haplotype resolution, high-depth sequencing of population genomes by mixing equimolar amounts of unlabeled individual DNA (Pool-seq) [20], and low-depth sequencing of multiple individuals in a population (lcWGR) [21,22]. High-depth sequencing is considered the gold standard for generating high-quality data that accurately identifies both coding and non-coding DNA variants but budgetary constraints and data-volume limitations often lead researchers to consider alternative approaches. Pool-seq offers cost-effective whole genome polymorphism data, though it is unsuitable for inferring haplotypes and linkage disequilibrium. lcWGR, with a typical depth of <1×, is suitable for large-scale population studies, balancing cost-effectiveness and high polymorphism density; however, reference genomes are relied upon for genotyping [23] and it is commonly used in genome-wide association studies (GWAS) [24]. In studies on pigs, lcWGR-based GWAS effectively screened for candidate genes (*CHD2*, *KATNAL2*, *SLC14A2*, and *ABCA1*) associated with reproductive traits [25]. Whole-exome genome sequencing (WEGS) has been proposed to explore peripheral arterial disease using high-depth whole-exome sequencing (WES) in combination with lcWGR, and the results suggest that the method can achieve a balance between efficiency, affordability, and coverage [26]. Advancements in sequencing technologies have enhanced technical support across scientific fields such as basic research and agricultural breeding, promoting genetic improvement and biodiversity conservation. As shown in Figure 1, the WGS workflow covers the entire process, from genomic sample preparation to data analysis, including steps such as DNA extraction, sequencing, reference genome alignment, and variant analysis.

### 2.2. Quality Control Parameters Related to WGS

The sequencing depth, coverage ratio, and mapping rate of WGS are essential metrics for assessing data quality and sequencing quantity [27]. Sequencing depth refers to the ratio of the total number of bases obtained by sequencing to the size of the genome, commonly denoted as “X” [28]. This metric significantly impacts the completeness and accuracy of de novo genome assembly [29], the number of genes detected in transcriptome sequencing and their expression levels [30], the ratio of rare variants to SNPs [31], and the accuracy of SNP calling and genotyping in whole genome sequencing [32]. Therefore, before performing WGS, the sequencing depth needs to be rationally selected to balance data quality and cost control [33].

Coverage refers to the proportion of sequenced regions relative to the entire target genome-specifically, the ratio of regions detected at least once compared to the total genome, expressed as a percentage (%) [28]. When detecting low-frequency mutations or performing detailed variant analysis, the higher the coverage and the lower the leakage rate, the higher the reliability and accuracy of sequencing. A positive correlation exists between sequencing depth and coverage; as sequencing depth increases, both error rates and false positives decrease. When the depth reaches a certain level, the coverage enters a saturation period, and the effect of further increasing sequencing depth on the coverage is no longer significant. Studies on WGS in pigs have shown that a depth of 10x can achieve effective genome coverage and accurately identify variant sites, with coverage exceeding 99% [34,35]. Conversely, at depths below 4×, false positive variants significantly increase, with coverage dropping to around 95% [36].

The mapping rate measures the proportion of bases in sequencing data that align to a reference genome, representing the percentage of matched bases relative to the total sequenced bases [37]. Higher mapping rates indicate better consistency between sequencing data and the reference genome, reflecting higher data quality. On the contrary, it means that the sample quality is low, the reference genome is incomplete, or there are problems such as repeated sequences [38]. Sequence alignment is key in the sequencing analysis workflow and requires efficient and accurate algorithm support [39,40]. In summary, sequencing depth and coverage ratio reflect the degree of genome-wide sequencing data coverage, determining the detection depth and accuracy, while the mapping rate provides insight into sequencing data reliability based on alignment with the reference genome.

### 2.3. Development and Iteration of Genome Sequencing Technologies

The first-generation sequencing technology, Sanger sequencing [41], offers high accuracy and medium read lengths; however, it has low throughput and relatively high costs [28]. Currently, genomics research also uses next-generation sequencing (NGS) [42] and third-generation sequencing (TGS) technologies [43]. NGS is noted for its speed and cost-effectiveness, generating large volumes of sequencing data [44] and contributing significantly to various genomic applications, including de novo sequencing [45], resequencing [46], transcriptomics [47], and epigenomics [48]. The Illumina sequencing platform remains the most widely used in NGS [49]. For example, researchers conducted a whole-genome sequencing study of methicillin-resistant Staphylococcus aureus (MRSA) in pigs from abattoirs in Cameroon and South Africa via Illumina sequencing, which showed that all isolates possessed multiple antibiotic resistance genes and six virulence factors [50]. The Illumina platform was applied to sequence the genomes of 57 goats from different agroforestry climates, generating DNA samples sequenced to an average depth of 9.71-fold, generating a total of about 2 TB of raw data and 24.76 million SNP loci, which is of great significance for understanding the genetic diversity and evolutionary dynamics of goats [51]. TGS technologies are characterized by ultra-long read lengths, high throughput, and improved accuracy; they primarily include single-molecule real-time (SMRT) sequencing [52] and synthetic long-read (SLR) sequencing [53], represented by platforms such as PacBio and ONT, which have greatly expanded the capabilities of genomic research [54,55]. ONT data can characterize SV, provide rapid pathogen identification, and assemble genomes [56]. The structural rearrangement in ASIP leading to deeper coat coloration was confirmed by GWAS analysis of coat coloration in Nellore bulls, and the SV with 1155 bp deletion and 150 bp transposon insertion was characterized using ONT data from 13 Australian Brahman heifers [57]. Furthermore, using ONT, 25 samples (15 Tibetan, 10 Han) were sequenced (average sequencing depth of 10×), and 27% more novel SVs were detected compared to NGS, constructing a more complete SV map of the Tibetan and Han populations’ genomes and screening 80 candidate genes for plateau environmental adaptation, which improves the understanding of the adaptation to multivariate loci [58]. TGS can more accurately resolve highly repetitive genomic regions and provide more precise haplotype construction, providing insight into population structure. Combining NGS and TGS technologies in WGS enables high-throughput, rapid, and precise data analysis, supporting in-depth analyses of complex genomes to address the diverse needs of biological research. Figure 2 shows the development timeline of sequencing technologies, while Table 1 presents a detailed comparison of the three sequencing generations.

### 2.4. Construction of High-Quality Reference Genomes

A reference genome is a standardized DNA sequence representing the “typical” genome of a species, rather than the complete DNA sequence of an individual. It serves as an idealized model, constructed by aggregating and optimizing genomic sequences from multiple individuals to capture the core genomic composition and structure of the species [59]. The use of high-quality reference genomes in population resequencing analyses improves the reliability of population genetic analyses, such as telomere-to-telomere (T2T) [60]. Since the emergence of the first draft of the human reference genome in 2000, a milestone in genome research [61], high-quality reference genomes have been accurately assembled from fragmented DNA sequences by increasing sequencing depth, using long-read sequencing technology, and optimizing assembly algorithms [62]. Long-read sequencing represented by PacBio HiFi and ONT solves the low-throughput limitation of Sanger sequencing and the short-read length constraint of NGS. High-quality reference genomes are the foundation of genomic research [63]. T2T achieves accuracy, continuity, and completeness of de novo genome assembly by combining multiple sequencing technologies such as PacBio HiFi, Hi-C, the triplex method, and ONT. In 2022, T2T successfully assembled the most comprehensive human reference genome for the first time [64]. T2T assembly generates a complete genome sequence, capturing comprehensive genetic information, particularly useful for complex genomes. Currently, T2T assembly has been applied to multiple species, as shown in Table 2.

## 3. Methods and Applications of Population Genomics Analysis Based on WGS

Population genomics integrates genomics and population genetics frameworks, aiming to explore genetic variations to reveal evolutionary pressures and unique genetic characteristics of species [79]. Population genomics clarifies the genetic structure, diversity, and ecological adaptation mechanism of species by analyzing the changes in gene or genotype frequencies caused by factors such as mutation, recombination, natural selection, genetic drift, and gene flow [80,81,82,83].

### 3.1. Inference of Population Genetic Structure

Population genetic structure analysis involves clustering subpopulations to assess structure and gene flow and infer historical evolution [83,84]. Inference methods are divided into parametric and non-parametric approaches [85]. Parametric methods, based on Hardy–Weinberg equilibrium (HWE) and linkage equilibrium (LE), utilize statistical models to determine structure and assign individuals to subpopulations. For effective HWE testing, one study developed a semi-supervised learning approach that estimates HWE by predicting common ancestors from genotypes, even when population structure and ancestry information are incomplete [86]. Commonly used models include the Wright–Fisher model and coalescent theory with extensions. The Wright–Fisher model assumes constant population size, random mating, and no mutation, gene flow, or selection, describing allele frequency changes within a population. In a study using this model to track single-locus allele frequencies through a mixed jump-diffusion process, a static distribution in mutation–selection–drift equilibrium was derived, revealing that mutations occur solely at monomorphic sites [87]. Coalescent theory, tracing genetic variation back to common ancestors, is flexible and applies to recombination, age structure, geographical structure, and population size changes [88]. STRUCTURE 2.3.4 and ADMIXTURE 1.3.0 are software applications used for population structure analysis, inferring genetic structure within populations and genetic origins of specimens from genotype data. Following model establishment, these tools use algorithms such as Approximate Bayesian Computation (ABC) [89], Maximum Likelihood Estimation (MLE) [90], and Markov Chain Monte Carlo (MCMC) [91], with ABC being the most common. A recent study presented the novel ONeSAMP 3.0 algorithm, which estimates effective population size (Ne) from the SNP data of endangered Channel Island fox populations using ABC and linear regression and validates the method using the Wright–Fisher hypothesis and empirical data [92]. STRUCTURE applies ABC and MCMC algorithms [93,94]. ADMIXTURE uses efficient MLE to infer ancestry, selecting population number (K) and analyzing known ancestry under supervised learning [95], it is more efficient with large datasets, providing an alternative to STRUCTURE due to its flexibility and speed [96]. ADMIXTURE analysis of cassava diversity identified five subpopulations, revealing extensive genetic variation among genotypes [97]. Studies using ADMIXTURE also traced domestication and genetic backgrounds of species like German White-headed Sheep [98] and dairy cows [99].

However, parametric methods may be limited by assumptions such as HWE and LE, which may affect the accuracy of their conclusions [100,101]. Subsequently, nonparametric methods emerged that did not rely on specific model assumptions [102]. Principal component analysis (PCA) is a widely used non-parametric method for inferring population structure and identifying outliers through dimensionality reduction. PCA simplifies high-dimensional data by identifying principal components (PCs), aiding in visualization, and revealing genetic patterns even in large datasets [103,104]. EIGENSTRAT and SmartPCA are common PCA tools for detecting structure, using eigenvalues to capture variance among individuals and eigenvectors to represent principal axes [105]. For example, the genomic analysis of calf pregnancies using EIGENSTRAT identified candidate genes associated with fertility (ASIC2 and SPACA3) [106]. Similarly, an analysis of bovine respiratory disease complex (BRDC) revealed regions associated with BRDC susceptibility [107]. Overall, PCA provides evidence for analyzing structure and gene flow [108], processing genotype datasets rapidly without complex parameters; however, robustness and repeatability are questionable [109].

Phylogenetic Analysis is essential for studying species diversification, exploring historical diversification contexts, and geographical distribution constraints. Phylogenetic reconstruction methods are divided into distance-based (e.g., Neighbor-Joining, NJ) and character-based approaches (e.g., maximum parsimony, maximum likelihood, and Bayesian inference). Software for constructing phylogenetic trees, such as MrBayes 3.2.2, MEGA 11.0, and PhyML 3.3, is often utilized in estimating the optimal number of subpopulations [110,111]. This estimation typically involves parametric and non-parametric methods, alongside phylogenetic approaches that leverage tree-building software to analyze evolutionary relationships and genetic diversity within populations. For example, HWE testing on Tobet dogs from Kazakhstan and Mongolia showed no significant frequency deviation, while STRUCTURE analysis revealed seven genetic clusters. NJ and maximum likelihood results suggested that the Tobet breed shares genetic traits with other breeds [112]. In a study on Chinese sheep breeds, whole-genome SNPs and InDels revealed phylogenetic relationships among eleven local and two foreign breeds, quantifying homozygosity [113]. Research on Qinchuan cattle subpopulations through WGS, NJ trees, and PCA showed QNC and ZSC with similar ancestry, while QCC was distinct, highlighting structure, differentiation, and diversity patterns [114].

In summary, analytical methods for studying population structure elucidate genetic relationships and ancestral histories, enhancing our understanding of genetic variation and species evolution. Table 3 outlines the strengths and limitations of these tools, guiding future research and methodological decisions.

### 3.2. Genomic Variation and Detecting Natural Selection

Genomic variation refers to a variety of changes that occur in the DNA sequence that can significantly affect the phenotype and fitness of an organism. Genomic variation consists mainly of SNPs, InDels, CNVs, and SVs [18,19]. The number and function of these variants can directly affect the survival and reproduction ability of individuals, thus providing the basic material for natural selection. In the analysis of WGS data, two main types of variants were focused on: SNP and InDel. The most commonly used tools for variant detection are Bcftools mpileup and the HaplotypeCaller module in the GATK 4.2.6.0 software [120]. In addition, the Atlas2 Suite employs logistic regression modeling and allows user-adjustable thresholds to accurately distinguish variants such as SNPs and InDel from sequencing and mapping errors with high sensitivity (96.7%) [121]. Natural selection drives species adaptation and evolution by selecting favorable variation; so, after determining the genetic structure and ancestral components of a population, the next step is to use selective scanning methods to detect regions of the genome that have been affected by natural selection, thereby identifying candidate genes associated with economic traits or environmental adaptations. Selective scanning identifies genes under selection by analyzing reduced diversity in regions surrounding selected loci. This is combined with metrics like nucleotide diversity (θπ) [122] and Tajima’s D [123] to reduce confounding effects in population statistics [124]. In regions near selected alleles, positive selection increases linkage disequilibrium, leading to observable extended haplotype homogeneity of varying lengths in the genome. Based on the genomic variation patterns, the assay signals were categorized by selscan 1.1.0 software [125] into an Extended Haplotype Homozygosity (EHH) test [126], Haplotype Homozygosity Score (iHS), and Cross Population Extended Haplotype Homozygosity (XP-EHH) test. Additionally, the composite likelihood ratio (CLR) test is equally applicable for long-term equilibrium selection tests [127], and others have further extended the CLR, i.e., the cross-population composite likelihood ratio (XP-CLR), which refers to a test that does not depend on changes in population size and simulates the differences in allele frequencies at multiple loci between two populations [128].

Through the above methods, WGS is suitable for describing the genetic characteristics of research populations in the study of population genomics on the one hand, thereby elucidating the biological mechanisms and physiological functions of these populations on the other. On the other hand, it is widely used in medicine to further understand the genetic basis of human health and disease as well as other traits. One study used high-depth WGS to explore the association between genomic variation and Parkinson’s disease (PD), identifying a total of 29,561 SVs, 32,153 CNVs, and 174,905 STRs, and found that CNV deletions were significantly enriched in PD autosomal end ratios, revealing a complex relationship between specific genetic variants and disease phenotypes [129]. In the first study of genome-wide long read-length resequencing and SV associated with phenotype, disease, and population adaptation in the Chinese Han population, 1929 loss-of-function SVs affecting the coding sequences of 1681 genes were annotated and revealed complex SV in the human genome [130]. In a study assessing genotypes of Altai sheep from China and three Kazakhstan sheep breeds, Fst and XP-EHH identified six candidate genes associated with reproductive traits: *ESR1*, *OXTR*, *MAPK1*, *RYR1*, *PDIA4*, and *CYP19A1* [131]. Similarly, in Yunnan semi-fine wool sheep, *FSHR*, *BMPR1B*, and *OXT* genes related to litter traits were identified through WGS and analyzed using Tajima’s D and iHS methods [132]. In research on the genomic diversity and selection characteristics of Xia nan cattle, 42 genes (θπ and CLR) and 131 genes (FST and XP-EHH) were detected, identifying a region with strong selection signals on BTA8. This study clarified Xia’nan cattle genomic characteristics, population structure, and selection signals related to economic traits [133]. In conclusion, selective scanning not only evaluates genetic diversity and population structure but also aids in identifying genes linked to adaptive traits or economic characteristics, providing essential support for livestock genetic improvement and crop breeding. Table 4 summarizes the advantages and disadvantages of various selective sweep detection methods and their applicable populations.

### 3.3. Detecting Localized Features of Genomic Evolutionary Processes

The classic neutral theory in modern evolutionary genetics posits that most molecular changes result from genetic drift, with positive selection playing a minor role. However, recent evidence shows that natural selection is prevalent in many genomes, significantly impacting genetic diversity across genomes. The Huttley–Kreitman–Aguadé (HKA) [138] and McDonald–Kreitman (MK) [139] tests are essential tools in studying molecular evolution and assessing natural selection’s effects on genetic variation. The HKA test evaluates the impact of variation patterns under the neutral evolution hypothesis by comparing the level of variation in a gene region within and between species [138]. This test is versatile, suitable for analyzing species with diverse evolutionary histories and traits, and considers intra- and inter-species variation, helping reveal selection and neutral evolution’s relationship [140]. The MK test focuses on the ratio of synonymous to non-synonymous mutations (dn/ds, KA/KS, or ω), inferring whether a mutation has undergone a natural selection and identifying negative (ω < 1) or positive (ω > 1) selection in genes [141]. A high number of non-synonymous mutations indicates positive or balanced selection in specific environments; in contrast, synonymous mutations typically undergo random drift without selection effects. For instance, 21 candidate genes for salt–alkali adaptation in the Amur carp (Leuciscus waleckii) were identified through genome-wide SNP screening, excluding geographic and environmental variations [142]. Additionally, four missense mutations in the SPTBN5 gene were identified by whole genome sequencing to be associated with hearing differences between zebu and common cattle [143]. In summary, methods like HKA and MK tests assess selection pressures on genetic variation, identifying selection patterns in specific genes or regions.

### 3.4. Analysis of Gene Penetration for Specific Environmental Adaptations

Gene penetration, or introgressive hybridization, refers to transferring genetic components from one population to another through hybridization and subsequent backcrossing between populations [144]. After identifying selection regions, gene flow analysis explores adaptive gene penetration into different populations, enhancing populations’ adaptive capacities by introducing genetic variation. Hybridization-driven gene flow shapes biodiversity and adaptation, necessitating advanced methods to quantify introgression [145,146]. Current methods, based on genetic similarities and allele frequency differences, include Patterson’s D statistic, chain disequilibrium, S* statistic, and probabilistic models (e.g., Hidden Markov Models and machine learning approaches) [147]. The D statistic (AB-BA-BABA test) is widely used to identify gene flow by comparing derived allele proportions between two sister species and a third species, inferring gene penetration amid incomplete lineage sorting [148]. For example, genome analysis of the Kouprey (*Bos sauveli*) showed similarities with *Bos javanicus* and *Bos gaurus*, while approximately 10% of Chinese zebu ancestry originates from mixed-population gene flow [149]. In a study on pigmentation in Lanping Black-Boned Sheep (LPB), combined genomic and transcriptomic data, along with a D statistic analysis of LPB and common sheep (LPN), traced LPB’s origin to LPN, identifying *ERBB4* and *ROR1* as pigmentation-associated candidate genes, revealing genetic backgrounds and variations related to LPB [150]. A study on the Murciano Granadina goat’s genetic background found that genetic penetrance likely originated in Morocco, with significant individual variation, suggesting Morocco as the breed’s origin and high genetic diversity in individual goats [151]. In summary, genetic penetration and variation across populations are crucial in biological evolution, deepening the understanding of genetic relationships among species and offering valuable insights for managing and utilizing biological resources.

Combining population genomics approaches with T2T assembly not only improves data processing and variant identification, but also provides insights into population structure, selection signals, and evolutionary relationships to support biodiversity, genetic improvement, and conservation efforts. Figure 3 summarizes the analysis methods discussed.

## 4. Summary and Outlook

Population genomics based on whole-genome sequencing (WGS) has made significant contributions to uncovering population structure, genomic variation, natural selection, and gene flow. Techniques such as principal component analysis (PCA) and ADMIXTURE have elucidated complex ancestral relationships, while methods like iHS and XP-EHH have identified genes associated with economically important traits. These insights have greatly advanced the genetic improvement of crops and livestock. However, despite the rapid accumulation of resequencing data, the current analytical methods face limitations, particularly in fully harnessing the data’s potential to provide a deeper understanding of genetic variation and evolution. Therefore, developing more comprehensive analytical frameworks and robust computational tools is crucial for maximizing the value of resequencing data.

The emergence of artificial intelligence (AI) and machine learning (ML), particularly deep learning (DL), is currently revolutionizing biological data analysis. These technologies enable the efficient processing of high-dimensional resequencing data, the identification of key genetic variations, and improved analytical accuracy. However, challenges related to interpretability and reproducibility persist, limiting their broader application in scientific research. Thus, the development of more interpretable and transparent models will be pivotal for future breakthroughs. Whole-genome resequencing provides a solid foundation for understanding genetic diversity and biological adaptability, and future advancements in this field will rely on technological innovations and interdisciplinary collaboration.

## Figures and Tables

**Figure 1 ijms-26-00372-f001:**
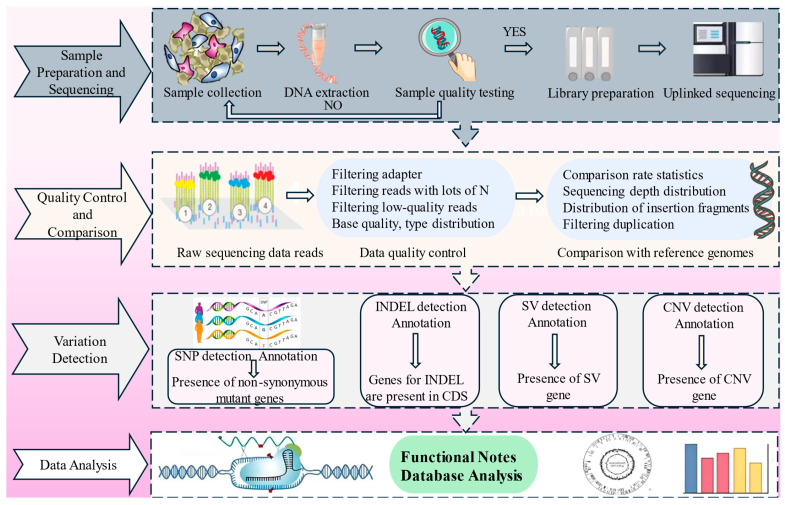
WGS workflow diagram.

**Figure 2 ijms-26-00372-f002:**
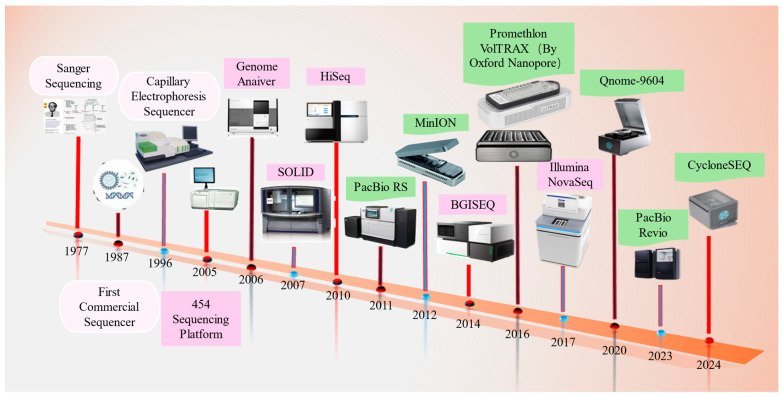
Evolution of sequencing technology. Note: pink boxes are for first-generation sequencing technology, purple boxes are for second-generation sequencing technology, and green boxes are for third-generation sequencing technology.

**Figure 3 ijms-26-00372-f003:**
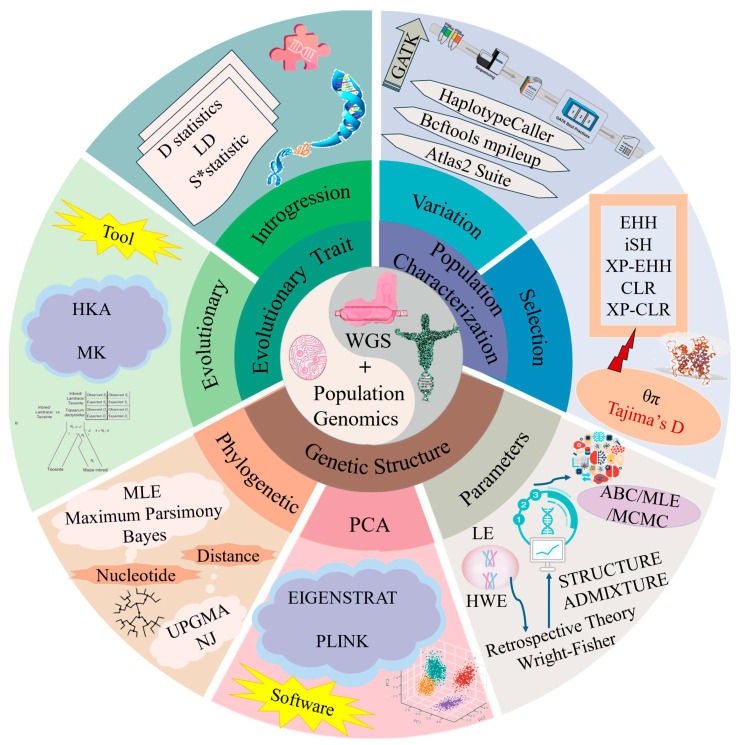
Population genomics analysis methods for whole genome sequencing data. Note: The second circle indicates the three domains of evolutionary traits, genetic structure, and population characteristics for the joint analysis of WGS and population genomics; the third circle indicates the analysis methods for each of the three domains; and the fourth circle indicates the tools or software used for the analysis methods.

**Table 1 ijms-26-00372-t001:** Comparison of three sequencing technologies.

Criteria	Sanger	NGS	TGS
Read Length	500–1000 bp	100–600 bp	10^3^–10^5^ bp
Throughput	Low	High	Moderate (capable of longer reads)
Cost	High	Low (large-scale sequencing is more cost-effective)	High
Sequencing Accuracy	High (>99.99%)	High (around 99%)	High initial error rates, improving over time
Application Scenarios	Small fragment or single-gene analysis	Large-scale genome and transcriptome sequencing	Complex regions, epigenetic studies
Platforms	ABI 3700	Illumina, Ion Torrent	PacBio, Oxford Nanopore

**Table 2 ijms-26-00372-t002:** T2T genomic information of some representative species.

Species	Genome Size	Sequencing Methods	Reference
Maize	2.16 Gb	CLR, ONT, Bionano, Hi-C	[65]
Fish	600.1 Mb	HiFi, Hi-C, ONT, trio	[66]
Rice	391 Mb	HiFi, CLR, Genome Matching	[67]
Barley	4.2 Gb	HiFi, Bionano	[68]
Human	3 Gb	HiFi, Hi-C, ONT, Bionano	[64]
Chicken	1.2 Gb	HiFi, ONT, trio	[69]
Mongolian gerbil	2.69 Gb	HiFi, Hi-C, ONT	[70]
Silkworm	449–468 Mb	HiFi, CLR, ONT	[71]
Carrot	427.33 Mb	HiFi, Hi-C, ONT	[72]
Kiwifruit	633 Mb		[73]
East Friesian sheep	2.96 Gb		[74]
Goose	106.89 Mb		[75]
African catfish	969.62 Mb		[76]
Sheep	2.85 Gb	HiFi, Hi-C, ONT, Bionano	[77]
Sorghum	724.85 Mb	HiFi, Hi-C, ONT	[78]

**Table 3 ijms-26-00372-t003:** Comparison of tools for population genetic structure analysis.

Methods	Software	Advantages	Disadvantages	References
Parametric Method	ADMIXTURE 1.3.0	Efficient processing of large-scale genomic data, estimating the ancestral components of individuals, suitable for multi-population analysis	No explicit consideration of linkage disequilibrium (LD) between markers, a priori K selection required	[95,96]
	STRUCTURE 2.3.4	Handles different types of markers for easy and flexible analysis	Longer computational time, model selection dependent on prior distribution	[94]
PCA	EIGENSTRAT 6.0.1	Easy-to-handle large-scale data and can use PCA to adjust genetic structures	Algorithms depend on data quality and sample selection	[115]
	PLINK 2.0	Efficient processing and analysis of genotype data with multiple statistical tests	Designed for SNP data only (GWAS output), limited support for other data types	[116]
	SmartPCA 1.0	PCA for large-scale datasets with easy visualization of results	Required for use with other software	[117]
Phylogenetic Tree	Mrbayes 3.2.2	Efficient extrapolation capabilities, support for multiple models, and reliable analytical results	Longer computation times and high memory requirements	[118]
	MEGA 11.0	Visualization features to support multiple evolutionary models	Less efficient and less flexible	[110]
	PhyML 3.3	Supports multiple evolutionary models with high accuracy for small-to-medium data sets	Computation is relatively slow, especially on large datasets	[119]

**Table 4 ijms-26-00372-t004:** Comparison of Key Genomic Selection Detection Methods.

Method	Principle	Advantages	Disadvantages	Applicable Populations
EHH [134]	Determine whether the frequency of a haplotype is skewed toward a particular variant by measuring the variability of the haplotype around that particular variant in the genome	Efficient at detecting recent selection signals, analyzes selection intensity through Extended Haplotype Homozygosity	Only detects recent positive selection, unable to identify negative or neutral selection	Detecting recent selection in a single population
iHS [135]	Assessment of selection pressure by comparing EHH values of variants under benign and adaptive selection	Capable of detecting incomplete selective sweeps, handles moderate selection signals, is a widely used tool	Insensitive to ancient selection, requires adjustment of detection windows for different datasets	Detecting selective sweeps within a single population, especially incomplete sweeps
XP-EHH [136]	Detection of selection signals by comparing differences between EHH values in a population and those in a reference population	Effectively identifies selective sweeps between populations, suitable for detecting long haplotype selection signals, handles regions with increased linkage disequilibrium	More sensitive to recent selection, weaker for ancient selection signals	Cross-population comparisons, especially in regions with high linkage disequilibrium
CLR [137]	Evidence of selection was determined by comparing the likelihood ratios between the observed sequence variation and the expected variation (assuming no selection)	Suitable for long-term balancing selection detection, can identify selective sweeps across the entire genome, applicable to different populations	Weaker at detecting complex selection signals, may have biases in hotspot regions	Whole-genome scans, particularly for selective studies across multiple populations
XP-CLR [128]	An extension of CLR to detect selection signals by comparing CLRs of different populations	Capable of comparing multi-locus allele frequency differences between two populations, suitable for detecting selective sweeps between populations	Does not account for population size changes, computationally intensive, long processing time	Comparisons between two different populations, e.g., Northern and Southern European populations

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
