# Peer review of "Advances in Whole Genome Sequencing: Methods, Tools, and Applications in Population Genomics"

_ijms, 2025, doi:10.3390/ijms26010372_

Round 1
Reviewer 1 Report
Comments and Suggestions for Authors
This is a very well written and educational paper about advances in Genome resequencing in animal population and I have no major questions or issues to address for revision. The pictures are well made and the different tables are illustrative.
There is some minor mistakes on line 299 (is should be deleted?), and 274 (. should be a ,)
Author Response
Dear Reviewer,
We are grateful for the opportunity to revise our manuscript titled " Advances in Whole Genome Sequencing: Methods, Tools, and Applications in Population Genomics " (Manuscript ID: ijms-3343660). Regarding the minor error you mentioned, we have fixed it accordingly:
The reviewer’s comment:
This is a very well written and educational paper about advances in Genome resequencing in animal population and I have no major questions or issues to address for revision. The pictures are well made and the different tables are illustrative.
There is some minor mistakes on line 299 (is should be deleted?), and 274 (. should be a ,)
Response:
Thank you for your positive feedback and detailed comments on our paper. We are pleased that you found the paper educational in terms of advances in genome resequencing in animal populations and recognized the quality of the illustrations and tables. We have revised the paper according to your suggestions, thank you again for your valuable comments, and we look forward to a revised manuscript that better meets your expectations.
Yours sincerely,
Dr. Weidong Deng
Yunnan Provincial Key Laboratory of Animal Nutrition and Feed, Faculty of Animal Science and Technology, Yunnan Agricultural University, Kunming 650201, China
Tel.: +86-871-65220375
Email: dengwd@ynau.edu.cn

Reviewer 2 Report
Comments and Suggestions for Authors
This is a relatively comprehensive review on whole genome sequencing technologies, analytical tools and applications in population genomic research. The review is well written and covers extensive knowledge, from basic concept to advanced technologies/methodologies in the related fields. I have some major concerns:
1. Current version of the review could only be used an educational materials for people unknown to the subject and field. In order to increase its novelty and insights for people working in the same field, some innovative ideas and insightful comments should be provided from the authors, eg, more elaboration on how the 3rd generation of long-read sequencing technologies could further facilitate population genomic research, what could be detected with TGS that NGS could not offer in terms of population structure, natural selection, environmental adaption, evolutionary history and introgression etc.?
2. in section 3.2, entitled "detecting natural selection and genomic variation", but with almost no description of genomic variation. In this entire review, it would be quite helpful if there is a dedicated section for genomic variation, where to give more details about types, quantities and functions, as well as tools to detect all genomic variations since different types of genomic variations would be used in different population genomic analyses.
3. Related to genomic variations, there have been two significant overlook in terms of population genomic research, as the review trying to cover comprehensively: (1) one major application of WGR in population genomics is to genetically characterize the population under study, in addition to population structure, selection, evolution, and adaption, WGR data from a population could also provide biological mechanism of many physical and functional traits of the current population; (2) another major, probably most significant application of WGR is in genomic medicine, ie, to better understand the genetic basis of human health and illnesses plus other traits. Even though genomic medicine would be beyond the scope of this review, but this aspect should be mentioned.
Some minor comments:
1. Even though traditionally, such approach was defined as resequencing, currently it's most common to use "whole genome sequencing, WGS" instead. So I suggest to use WGS instead of WGR, better for later key word searching etc..
2. in section 2.2 "Indicators related to whole genome resequencing ", it's better to use "quality control parameters"?
3. In Figure 3, the second layer of circle, in addition to evolutionary traits and population structure, a 3rd component of "population characterization" or something equivalent could be added, corresponding to my above-mentioned "genomic variation" part; the 3rd layer of circles should be unified with parallel approaches and labelled accordingly; the 4th circles could be various exemplary tools and programs.
Comments on the Quality of English LanguageSome terms need to be more precise.
Author Response
Dear Reviewer,
We are grateful for the opportunity to revise our manuscript titled " Advances in Whole Genome Sequencing: Methods, Tools, and Applications in Population Genomics " (Manuscript ID: ijms-3343660). We appreciate the insightful comments and suggestions provided by the reviewers, which have significantly improved the quality of our work.
In response to the reviewers' feedback, we have made several key revisions to enhance the clarity, specificity, and impact of our manuscript:
The reviewer’s comment 1:
This is a relatively comprehensive review on whole genome sequencing technologies, analytical tools and applications in population genomic research. The review is well written and covers extensive knowledge, from basic concept to advanced technologies/methodologies in the related fields. I have some major concerns:
Current version of the review could only be used an educational materials for people unknown to the subject and field. In order to increase its novelty and insights for people working in the same field, some innovative ideas and insightful comments should be provided from the authors, eg, more elaboration on how the 3rd generation of long-read sequencing technologies could further facilitate population genomic research, what could be detected with TGS that NGS could not offer in terms of population structure, natural selection, environmental adaption, evolutionary history and introgression etc.?
Response 1:
Thank you very much for your suggestion. Based on your suggestions, we have revised and supplemented the article accordingly to enhance its novelty and inspiration for researchers in related fields. We have added a discussion on how third generation sequencing (TGS) technology can further facilitate population genome research to the paper, emphasizing the advantages of TGS. Specific study cases are added to the paper to demonstrate how TGS can provide richer information than NGS in practical applications. A more in-depth analysis of the comparison between NGS and TGS in resolving highly repetitive genomic regions and constructing haplotypes emphasizes the advantages of the two technologies when used in combination in the analysis of complex genomes.
The reviewer’s comment 2:
In section 3.2, entitled "detecting natural selection and genomic variation", but with almost no description of genomic variation. In this entire review, it would be quite helpful if there is a dedicated section for genomic variation, where to give more details about types, quantities and functions, as well as tools to detect all genomic variations since different types of genomic variations would be used in different population genomic analyses.
Response 2:
Thank you for your careful review and pertinent feedback on our paper. We value your comments on section 3.2 entitled “Detection of natural selection and genomic variation” and realize that the description of genomic variation therein is inadequate. To address this issue, we have added content related to genomic variation, introducing in turn the definition and importance of genomic variation, the basic idea of variation detection, and a discussion of related tools and methods. For additions, see lines 370 to 380.Thank you again for your comments and we look forward to your further feedback on the revised paper.
The reviewer’s comment 3:
Related to genomic variations, there have been two significant overlook in terms of population genomic research, as the review trying to cover comprehensively: (1) one major application of WGR in population genomics is to genetically characterize the population under study, in addition to population structure, selection, evolution, and adaption, WGR data from a population could also provide biological mechanism of many physical and functional traits of the current population; (2) another major, probably most significant application of WGR is in genomic medicine, ie, to better understand the genetic basis of human health and illnesses plus other traits. Even though genomic medicine would be beyond the scope of this review, but this aspect should be mentioned.
Response 3:
Thank you for your valuable suggestions. Following your suggestions, we have emphasized in the paper that WGR data can not only reveal population structure, selection, evolution, and adaptation, but also provide a biomechanistic understanding of many physical and functional traits in current populations. Several studies on genomics were cited in the revision to better understand the potential of WGR in human health and disease-related genetic variation and its importance in the study of other traits. These studies not only support the mining of candidate genes for population-related traits, but also emphasize the importance of genomic variation in medical research. We hope that these revisions will enhance the academic depth and breadth of the paper and make it more attractive and informative. Thank you again for your valuable comments and we look forward to your further feedback.
Minor revision 1:
Even though traditionally, such approach was defined as resequencing, currently it's most common to use "whole genome sequencing, WGS" instead. So I suggest to use WGS instead of WGR, better for later key word searching etc.
Response 1:
We agree with your suggestion to use “whole genome sequencing (WGS)” instead of “whole genome resequencing (WGR)”. We will modify the paper accordingly to ensure consistency of terminology and to improve the effectiveness of keyword searches.
Minor revision 2:
In section 2.2 "Indicators related to whole genome resequencing ", it's better to use "quality control parameters"?
Response 2:
We recognize your suggestion to use “quality control parameters” in section 2.2, “Metrics related to whole genome resequencing”. We have revised this section to reflect this more accurate terminology.
Minor revision 3:
In Figure 3, the second layer of circle, in addition to evolutionary traits and population structure, a 3rd component of "population characterization" or something equivalent could be added, corresponding to my above-mentioned "genomic variation" part; the 3rd layer of circles should be unified with parallel approaches and labelled accordingly; the 4th circles could be various exemplary tools and programs.
Response 3:
We thank you for your suggestion for Figure 3. We have added “Population Characterization” as the third component of the second layer of circles to correspond to the “Genomic Variation” section you mentioned. At the same time, we will harmonize the labeling of the third layer of circles and ensure consistency with the parallel approach. In addition, we will add various example tools and programs to the fourth circle.
Yours sincerely,
Dr. Weidong Deng
Yunnan Provincial Key Laboratory of Animal Nutrition and Feed, Faculty of Animal Science and Technology, Yunnan Agricultural University, Kunming 650201, China
Tel.: +86-871-65220375
Email: dengwd@ynau.edu.cn

Reviewer 3 Report
Comments and Suggestions for Authors
The authors have conducted a nice review of the whole genome resequencing. I do appreciate the effort and time however, the review is not well composed with much misinformation and no attention to details. Additionally, they state in the aim for significant goals that will be achieved, such as “theoretical references for demographic inference based on population genomics and for studying evolutionary” which I am afraid the review cannot deliver. Figures show effort, but the layout looks not optimal and they contain miss accuracies. If the review can be re-written with new figures and information that promotes the field and has solid background information, then it can be of interest. Apologies, I stopped correcting the meaning after some point.
Introduction
Fist sentence is not correct, does not only count for genes? How about the study of non-coding regions this is not genomics?
Next sentence (L30) does not match the meaning of the previous one.
L31 is identical for L28, rephrase please
L40 why sheep is a complicated genome?
L40 Whole genome resequencing term needs definition.
Introduction is not well written. Misses good flow and does not demonstrate the need for a review while the aim mentions “theoretical references for demographic inference based on population genomics and for studying evolutionary and domestication mechanisms of economically important crops and livestock, as well as practical guidance for processing large-scale resequencing datasets.” Which was not before explained in the next.
Rest Manuscript
L80 I do not agree with the terms low coverage and high coverage. High above 50x it is not correct and it is an arbitrary threshold. Rephrase those definitions
Figure 1 has many issues. It is not accurate information regarding genomics analysis, how is functional analysis is a result of SNPs? I do not understand uplink sequencing and more. I am not an emoji person so also smile and sad face it is not clear to me.
L104 What does it mean every base is read?
L105 Coverage is important, but it has a plateau it is not the metric to completely define those.
Author Response
Dear Reviewer,
We are grateful for the opportunity to revise our manuscript titled " Advances in Whole Genome Sequencing: Methods, Tools, and Applications in Population Genomics " (Manuscript ID: ijms-3343660). We appreciate the insightful comments and suggestions provided by the reviewers, which have significantly improved the quality of our work.
In response to the reviewers' feedback, we have made several key revisions to enhance the clarity, specificity, and impact of our manuscript:
The reviewer’s comment 1:
The authors have conducted a nice review of the whole genome resequencing. I do appreciate the effort and time. However, the review is not well composed with much misinformation and no attention to details. Additionally, they state in the aim for significant goals that will be achieved, such as “theoretical references for demographic inference based on population genomics and for studying evolutionary” which I am afraid the review cannot deliver. Figures show effort, but the layout looks not optimal and they contain miss accuracies. If the review can be re-written with new figures and information that promotes the field and has solid background information, then it can be of interest. Apologies, I stopped correcting the meaning after some point.
Response 1:
Thank you for your suggestions and feedback on our paper, which is a review aimed at exploring WGS methods and providing theoretical guidance for subsequent applied genomic research. Based on your suggestions, we realize that there is some inaccurate information and insufficient details in the article. We will thoroughly review the entire article (including after line 105) to ensure the accuracy and clarity of all information. In addition, we have revised the graphs and charts to improve the clarity and readability of the graph layout, to ensure the accuracy of all data and information, and have added relevant background information and literature to improve the depth and breadth of the paper. At the same time, we understand your concerns about the goals of the review. We will further clarify and concretize our objectives to ensure that the review content supports the theoretical references we propose and provides valuable insights for population genomics and evolutionary studies. Thank you again for your valuable comments and we hope that these improvements will make the paper more attractive and academically valuable. We look forward to your further feedback.
The reviewer’s comment 2:
Introduction
Fist sentence is not correct, does not only count for genes? How about the study of non-coding regions this is not genomics?
Next sentence (L30) does not match the meaning of the previous one.
L31 is identical for L28, rephrase please
Response 2:
Thank you for your detailed feedback on the introductory section of our article. We have made the necessary changes and refinements based on your suggestions to improve the quality and flow of the article. Our reference to “genome research” is indeed too narrow, and the statement at the beginning of the introduction has been modified to cover the entire genome research, including the noncoding regions, to clarify the broad definition of genomics in this article, and the paragraph has been revised and integrated to ensure logical consistency. Revised to read: “Genome is the collection of all genetic information of an organism, including all genes and their regulatory and non-coding regions. Genome sequencing reveals the genetic diversity of species, population structure, domestication processes, evolutionary and adaptive mechanisms, and identifies loci of variation associated with economically important traits”.
The reviewer’s comment 3:
L40 why sheep is a complicated genome?
Response 3:
We thank the reviewers for their comments. We refer to yellow cattle, yaks, bison, goats and sheep as complex species because they exhibit rich genetic variation and diversity involving multiple genes and their interactions on phenotype. Their genome structures are often complex, not only due to the diversity of genome sizes and repetitive sequences, but also due to the fact that they exhibit complex genetic adaptations during evolution and domestication. These factors make sequencing and analyzing the genomes of these species more challenging compared to traditional model organisms. Therefore, we believe that they do fall into the category of complex species.” But to avoid ambiguity, we've changed the sentence heel to read: “The scope of sequenced organisms has expanded from traditional model organisms to include economically important crops, livestock, and endangered plants, with suc-cessful assemblies achieved for species such as yellow cattle [3], yaks [4], zebu [5], goats [6], and sheep [7]”.
The reviewer’s comment 4:
L40 Whole genome resequencing term needs definition.
Response 4:
Thank you for your in-depth analysis and feedback on the introductory section of our article. We have added a definition of “whole genome sequencing” to the Introduction. Specifically, we will add the following: “Whole genome sequencing (WGS) refers to sequencing the entire genome of multiple individuals within a species using a known reference genome sequence.”
The reviewer’s comment 5:
Introduction is not well written. Misses good flow and does not demonstrate the need for a review while the aim mentions “theoretical references for demographic inference based on population genomics and for studying evolutionary and domestication mechanisms of economically important crops and livestock, as well as practical guidance for processing large-scale resequencing datasets.” Which was not before explained in the next.
Response 5:
Thank you for your in-depth review and constructive comments on our manuscript. We have reorganized the structure of the introduction so that it clearly introduces the topic of whole genome sequencing and enhances the flow of the article. A discussion on the importance of whole genome sequencing in understanding population genomics, evolution and domestication mechanisms has also been added to the introduction, and the objectives of the review have been rephrased. Thank you again for your feedback and support, and if you have any additional suggestions or need for further revisions, we would be happy to hear from you and look forward to your further guidance.
The reviewer’s comment 6:
Rest Manuscript
L80 I do not agree with the terms low coverage and high coverage. High above 50x it is not correct and it is an arbitrary threshold. Rephrase those definitions
Response 6:
Thank you for your detailed review of our manuscript and your helpful suggestions. We have revised the original text to address your reference to the use of the terms “low coverage” and “high coverage” and their thresholds. We have revised it to read: Based on sequencing depth, coverage, and sample size, WGS techniques can be categorized into [20]: individual sequencing with high depth of coverage based on differences in haplotype resolution, high-depth sequencing of population genomes by mixing equimolar amounts of unlabeled individual DNA (Pool-seq) [21], and low-depth sequencing of multiple individuals in a population (lcWGR) [22,23]”. We hope that this modification will eliminate the arbitrary specification of thresholds while more accurately distinguishing between different sequencing technologies. We appreciate your comments and look forward to your further feedback.
The reviewer’s comment 7:
Figure 1 has many issues. It is not accurate information regarding genomics analysis, how is functional analysis is a result of SNPs? I do not understand uplink sequencing and more. I am not an emoji person so also smile and sad face it is not clear to me.
Response 7:
Thank you for reviewing our paper and for your detailed feedback on Figure 1. While the diagram is intended to show the complete workflow of whole genome sequencing (WGS), including sample preparation and sequencing, quality control and comparison, variant detection, and data analysis, we are aware that the term “upstream sequencing” may be difficult to understand. Therefore, we have supplemented the diagrams and removed the emoticons in the diagrams to provide more detailed explanations to ensure clarity.
The reviewer’s comment 8:
L104 What does it mean every base is read?
Response 8:
Thank you for your careful review of our article and for your valuable comments. Regarding the expression “each base was read” in line 104, we have revised and clarified it. In order to express the concept of “sequencing depth” more clearly, we have revised it to read: “Sequencing depth refers to the ratio of the total number of bases obtained by sequencing to the size of the genome.”
The reviewer’s comment 9:
L105 Coverage is important, but it has a plateau it is not the metric to completely define those.
Response 9:
Thank you for your valuable comments on our paper. We fully understand your point of view and recognize that the original formulation may not be clear enough. We will reword the content of line 170 to more accurately describe this concept. We will revise it to read, “When the depth reaches a certain level, the coverage enters a saturation period, and the effect of further increasing sequencing depth on the coverage is no longer significant.”
Yours sincerely,
Dr. Weidong Deng
Yunnan Provincial Key Laboratory of Animal Nutrition and Feed, Faculty of Animal Science and Technology, Yunnan Agricultural University, Kunming 650201, China
Tel.: +86-871-65220375
Email: dengwd@ynau.edu.cn

Round 2
Reviewer 3 Report
Comments and Suggestions for Authors
The authors have done an great job addressing the comments and suggestions provided during the review. The revised manuscript is now well-organized, clear, and significantly improved in its overall quality. The thoughtful incorporation of feedback has strengthened the key messages and enhanced the study's impact. Thank you for your efforts and commitment to producing a polished and interestng manuscript.